# YOLOv7oSAR: A Lightweight High-Precision Ship Detection Model for SAR Images Based on the YOLOv7 Algorithm

Yilin Liu [1,2] , Yong Ma [1], Fu Chen [1,*], Erping Shang [1], Wutao Yao [1], Shuyan Zhang [1] and Jin Yang [1]

1. Aerospace Information Research Institute, Chinese Academy of Sciences, Beijing 100094, China; liuyilin21@mails.ucas.ac.cn (Y.L.); mayong@aircas.ac.cn (Y.M.); shangep@aircas.ac.cn (E.S.); yaowt@aircas.ac.cn (W.Y.); zhangshuyan17@mails.ucas.ac.cn (S.Z.); yangjin@aircas.ac.cn (J.Y.)
2. College of Resources and Environment, University of Chinese Academy of Sciences, Beijing 100049, China
* Correspondence: chenfu@aircas.ac.cn

**Abstract:** Researchers have explored various methods to fully exploit the all-weather characteristics of Synthetic aperture radar (SAR) images to achieve high-precision, real-time, computationally efficient, and easily deployable ship target detection models. These methods include Constant False Alarm Rate (CFAR) algorithms and deep learning approaches such as RCNN, YOLO, and SSD, among others. While these methods outperform traditional algorithms in SAR ship detection, challenges still exist in handling the arbitrary ship distributions and small target features in SAR remote sensing images. Existing models are complex, with a large number of parameters, hindering effective deployment. This paper introduces a YOLOv7 oriented bounding box SAR ship detection model (YOLOv7oSAR). The model employs a rotation box detection mechanism, uses the KLD loss function to enhance accuracy, and introduces a Bi-former attention mechanism to improve small target detection. By redesigning the network's width and depth and incorporating a lightweight P-ELAN structure, the model effectively reduces its size and computational requirements. The proposed model achieves high-precision detection results on the public RSDD dataset (94.8% offshore, 66.6% nearshore), and its generalization ability is validated on a custom dataset (94.2% overall detection accuracy).

**Keywords:** YOLO algorithm; ship detection; oriented bounding box; Synthetic aperture radar (SAR) images

## 1. Introduction

In coastal military and industrial operations, achieving the all-weather and precise detection of ship targets is imperative for the swift allocation of resources. This capability plays a pivotal role in upholding national maritime security and fostering civilian production activities. To address this objective, an array of prior studies have been conducted. Synthetic aperture radar (SAR) microwave imaging technology has garnered considerable attention, in contrast to conventional optical imaging techniques, given its all-weather, continuous, and anti-interference detection capabilities. With the ongoing advancement of microwave remote sensing technology, numerous satellites (such as Nuwa, Haise, Qilu, ICEYE, StriX, Capella, TerraSAR, and Radasat2) are equipped with SAR sensors, contributing to a substantial enhancement in SAR image resolution. However, existing SAR ship detection algorithms face challenges related to suboptimal precision and deployment convenience. Therefore, there is an urgent need to refine algorithmic accuracy and enhance their deployability in order to meet these critical objectives [1–4].

The methods for detecting ships based on SAR images are commonly categorized into two groups: traditional algorithms [5,6] and deep learning algorithms. Traditional algorithms include Constant False Alarm Rate (CFAR) methods, such as adaptive CFAR (e.g., VI-CFAR, VTM-CFAR) and ordered statistics CFAR (e.g., OS-CFAR, CMLD-CFAR) [6–9]. These algorithms exhibit poor detection accuracy, require manual clutter modeling, and

have limited generalization capabilities, making them less commonly used. With the rise of artificial intelligence, deep learning algorithms have become mainstream in target detection tasks. Notable algorithms include Regions with CNN features (RCNN) [10], Fast R-CNN [11], Faster R-CNN [12], You Only Look Once (YOLO) [13–18], SSD [19], and Retina-Net [20].

In the field of SAR image detection, algorithms initially applied horizontal bounding box-based methods for ship detection. The first improved Faster R-CNN algorithm, combining traditional and deep learning methods, was proposed in 2017, achieving satisfactory results [21]. Subsequent researchers, like Zhao, Y and Zhao, L et al., aiming to address the challenges of detecting multi-scale ships and complex backgrounds in SAR images, introduced innovative approaches. These include a two-stage detection network called the Attention Receptive Pyramid Network (ARPN), which incorporates receptive field modules (RFB) and convolutional block attention modules (CBAM) to establish a top-down fine-grained feature pyramid. This enhances the relationship between non-local features and refines information from different feature maps to improve the detection performance of multi-scale ships in SAR images [22]. To further enhance the multi-scale detection capability of models, various researchers introduced numerous multi-scale detection modules or modified network structures. For instance, Wei et al. introduced a High-Resolution Ship Detection Network (HR-SDNet) based on HR-net, which combines high-resolution and low-resolution convolutional layers to achieve accurate and robust ship detection [23]. Chen Zhuo, using YOLOv7, presented the CSD-YOLO algorithm for complex scenes multi-scale ship detection. This algorithm incorporates Shuffle Attention (SA) mechanisms and Atrous Spatial Pyramid Pooling (ASPP) to enhance the model's multi-scale feature extraction capabilities, demonstrating the adaptability of YOLOv7 to complex-background SAR ship detection [24]. Additionally, Jingpu Wang, aiming to address the low detection rates and high false positives in detecting small ships in SAR images, redesigned a feature extraction network based on the characteristics of the ship targets in SAR images. The proposed Path Argumentation Fusion Network (PAFN) improves the fusion of different feature maps. Although the aforementioned methods show promising performances in SAR ship detection, effectively capturing ship targets, the imaging mechanism of SAR images differs from optical images [25]. Due to the different imaging mechanisms between SAR and optical images, factors such as the speed, direction, size, and material of ships can affect the imaging results of SAR sensors, leading to blurred boundaries of ship targets in the images and presenting challenges for precise localization. Most algorithms are primarily based on traditional horizontal bounding boxes, making it difficult to accurately describe and locate targets that require specific orientation information, such as ships. Therefore, a more flexible approach is needed to adapt to the irregular shapes of ship targets in SAR images and the continuously changing orientation of these ships, providing accurate boundary box information for precise localization of the ship targets.

To address the issue of inaccurate localization using horizontal bounding boxes for ship detection in SAR images, many scholars have introduced the concept of rotated bounding boxes into SAR object detection models. Some researchers, such as Zicong Zhu et al., have adapted network architectures to meet the demands of rotated box detection. Zicong Zhu introduced a new representation called non-continuous angle representation, and designed an automatic organization mechanism and a specific head structure for organizing points, proposing the AutoAnchor Network, which focuses on the detection of rotated objects [26]. Jiaming Han, Jian Ding, and others pioneered the rotation-equivariant detector (Redet) to tackle the problem of objects distributed in arbitrary directions. This method introduced a rotation-equivariant network to precisely predict orientations and significantly reduce model size [27]. Additionally, they proposed rotation-invariant RoI Align (RiRoI Align), which adapts to the orientation of RoIs for feature extraction. Subsequently, they further proposed the single-stage alignment network (S2A-Net), including a feature alignment module (FAM) and an oriented detection module (ODM), which improved its detection performance of multi-scale ships in SAR images [28]. Other researchers directly improved

existing object detection networks to adapt to rotated bounding box detection scenarios. For example, Jian Ding, Nan Xue, and others proposed the ROI-Transformer model, which introduces spatial transformations into the Region of Interest (RoI) and learns transformation parameters under the supervision of oriented bounding box (OBB) annotations. This innovative approach effectively addresses challenges in computer vision related to aerial image object detection. In cases where OBB annotations are available, the ROI-Transformer outperforms deformable Position-Sensitive RoI pooling, highlighting its flexibility and effectiveness in improving detection accuracy and alignment [29]. Yongchao Xu and their team introduced the Gliding Vertex model, which accurately describes multi-oriented objects by sliding the vertex on the edge of a horizontal bounding box, avoiding the direct regression issues that may cause confusion. By introducing length ratio regression and an obliquity factor based on the area ratio, these additional target variables are added to the Faster R-CNN regression head, achieving minimal computational overhead [30]. Zheng Ziyang and others proposed a method for detecting rotating ships in Synthetic aperture radar (SAR) images, introducing the concept of rotation angles to make it suitable for rotated box detection scenarios. They also introduced a rotation box loss function and a transfer attention module, enhancing the performance of a YOLOv4-based rotating box target detection network in SAR image ship detection [31]. Ge Ji, Wang Chao, and others aimed to address the azimuth-sensitive object detection issue in complex scenes in SAR images, using aircraft detection in complex SAR scenes as an example. They adopted YOLOX as their basic framework, introducing an inverted pyramid convolution aggregation network and proposing a spatial orientation attention module. They also introduced a spatial orientation-enhanced path aggregation feature pyramid network to capture feature transformations in different directions, emphasizing object features and suppressing background effects. Building upon this, they proposed a network enhanced with spatial orientation attention [32]. These algorithms efficiently and accurately perform target detection in various application scenarios. However, for nearshore ship detection, due to large and complex scenes, dense distributions, and the arbitrary orientations of ship targets, the discrimination and regression of the above algorithms are not accurate enough. These algorithms have inadequately tackled certain challenges in the academic domain. Examples include the accurate measurement of predicted values versus ground truth for the boundaries of rotated boxes during the training process, the handling of errors in loss when detecting square-like objects, and the complexities associated with their intricate designs. Furthermore, the substantial increase in model parameters resulting from these complexities poses challenges in the deployment phase within academic research settings. Their complex designs also lead to large model parameters, making deployment challenging.

To tackle the challenges associated with complex models, which result in large parameters and high computational requirements, hindering deployment, researchers have explored alternative strategies. Some have undertaken studies to address these issues. For instance, Yang-Lang, Chang et al. focused on mitigating the problem of excessive information in remote sensing images, which makes training models challenging. They developed a streamlined YOLOv2 model, named YOLOv2-reduced, by eliminating unnecessary layers in the detection process of the You Only Look Once version 2 (YOLOv2) model. This approach aimed to reduce computation time, enhance GPU utilization, and improve detection accuracy [33]. Similarly, Xiaowo Xu et al. aimed to decrease model parameters while preserving detection accuracy. They opted for model pruning while adopting the lightweight YOLOv5s model [34]. Although these methods can address the issue of oversized models, the process of redesigning and pruning networks is relatively complex. Additionally, their effectiveness is influenced by the dataset, making them solutions that may not universally apply to the problem.

This paper introduces a novel rotation box object detection model called YOLOv7oSAR, aiming to address the challenges of low accuracy, the imprecise localization of small targets, and the hindrance to deployment posed by the complexity of current rotation box

algorithms. This model is based on the YOLOv7 [16], a continuation of the YOLO algorithm series released by the original authors. The algorithm demonstrates high-precision detection results on the Rotation box SAR Ship Detection Dataset (RSDD) published by Professor Xu Cong'an's team in 2022 [35]. To tackle the issues of arbitrary target orientation and challenging localization, this paper introduces a rotation box detection mechanism built upon the YOLOv7 framework.

The following are the contributions of this article:

1. The proposed YOLOV7oSAR is a highly accurate SAR image ship detection network based on the YOLOV7 framework. It introduces Kullback–Leibler Divergence (KLD) loss [36] during training to enhance the accuracy of the inference model's final results.
2. To improve the model's small target detection capability in large scenes, the article introduces the BRA dynamic sparse attention mechanism [37]. This is a plug-and-play BCBS structure, which, compared to traditional self-attention mechanisms, acquires critical information from the entire image more quickly with lower spatial complexity, thereby enhancing the model's capability to detect small targets.
3. In order to reduce model parameters, this article redesigns the network's depth and width to align with research objectives. Additionally, it introduces the Partial convolution (PConv) structure [38] and proposes the lightweight P-ELAN structure, ensuring a reduction in model parameters without significantly affecting model performance. This reduction in parameters contributes to a smaller model size and lower hardware computational requirements.

The subsequent sections of this article are structured as follows: Section 2 provides an introduction to the materials and methods, Section 3 outlines the experimental design, and Section 4 articulates the findings from ablation experiments, comparative experiments, and validation experiments, followed by a comprehensive discussion. Section 5 offers a summary of the entire article, concluding with insights into future prospects.

## 2. Materials and Methods

This section provides an overview of the entire research process, illustrating the technical roadmap, which is shown in Figure 1. This study involves enhancing the YOLOv7 network for ship detection in Synthetic aperture radar (SAR) images. Initially, the model is trained and improved on the publicly available RSDD. In the shallow regions of the model, a BRA attention mechanism is introduced to enhance its capability of detecting small targets. Additionally, the network architecture is modified by incorporating Pconv convolution to replace the ELAN structure, aiming to make the model more lightweight. Subsequently, to validate and test the model's detection performance in different scenarios, a dataset for detecting ships, with slanting bounding boxes, in a harbor is created.

Section 2.1 specifically outlines the structure of the baseline network, YOLOv7. In Section 2.2, the structure of the YOLOv7oSAR model is introduced. Additionally, Section 2.3 covers the introduction of the small target detection module and lightweight detection module.

### 2.1. YOLOv7 Network Architecture

In the field of real-time object detection, the YOLO series is highly acclaimed for its remarkable balance between computational efficiency and accuracy. YOLOv7, released by the original team with practical paper support, is one of the most recent versions of the recently released YOLO algorithms. This paper focuses on improving the YOLOv7 algorithm. Its network comprises a backbone and a head, with input image dimensions set at $512 \times 512$. The head generates three layers of feature maps and introduces the BCBS structure, consisting of the Convolutional-Batch Normalization-Silu (CBS) and Convolutional-Batch Normalization-Sigmoid (CBM) layers, and incorporating a BRA attention mechanism. As SAR images typically have a single channel, preprocessing involves converting them into a three-channel format. Feature extraction begins with four layers of CBS, followed by MP-1 and ELAN layers, generating feature maps at different scales. These feature maps

are input into the head network for further processing, where the ELAN layer enhances feature extraction efficiency and robustness by regulating long and short gradient paths. The model introduces the PConv layer for lightweighting [16].

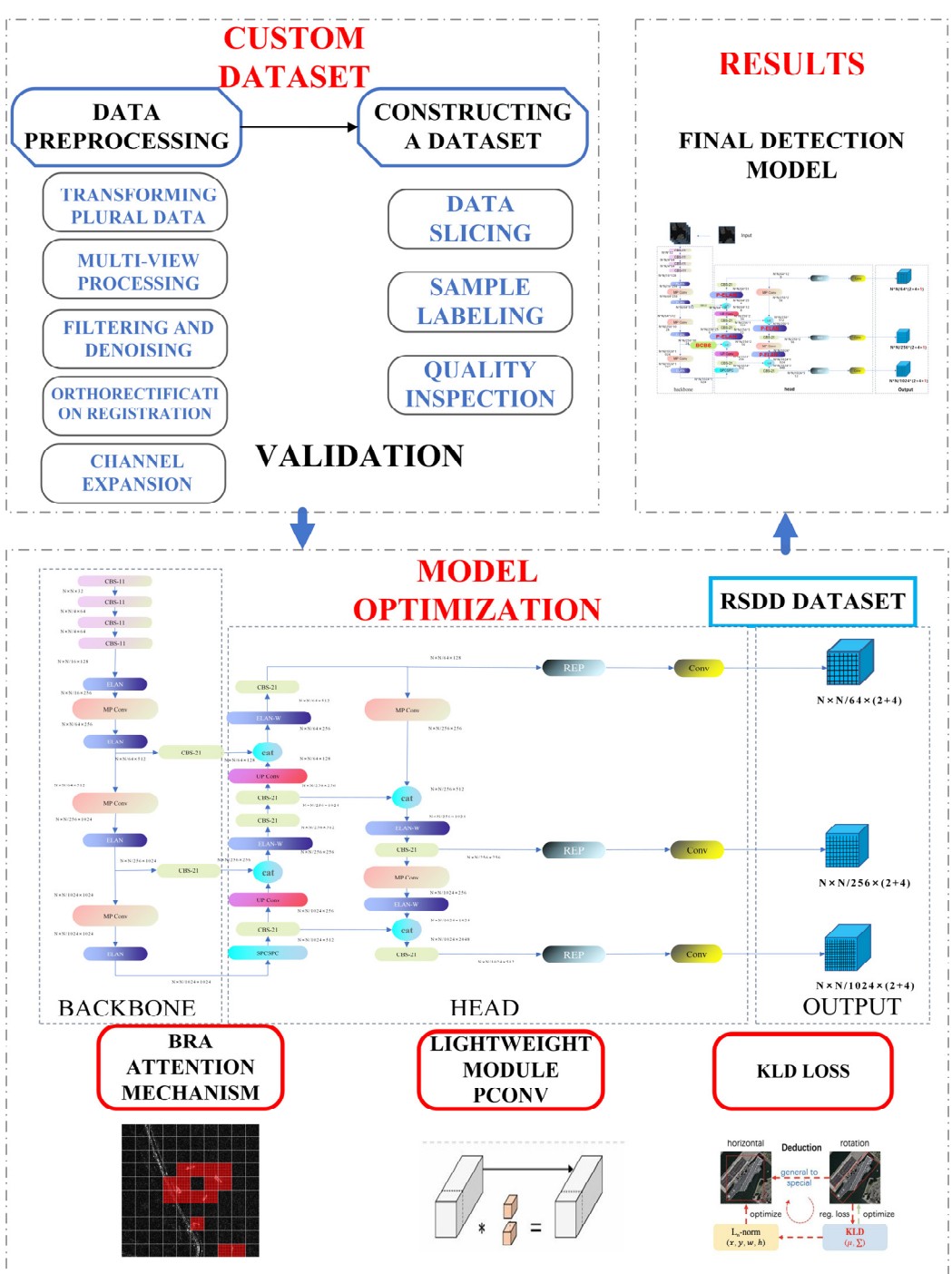

**Figure 1.** Technical roadmap of this study (The asterisk (*) represents the multiplication operation).

The improvements in YOLOv7 are mainly manifested in the ELAN structure, reparameterized model architecture, and effective label assignment strategy. These enhancements elevate its detection performance while reducing the model parameters. Utilizing this model for ship target detection in SAR images ensures heightened accuracy and significantly reduces the required hardware performance. This nuanced balance makes the

model particularly practical in scenarios with limited computational resources, all while maintaining exceptional detection efficacy.

### 2.2. YOLOv7oSAR Network Architecture

To achieve a more precise localization of ship targets, this paper introduces rotated bounding boxes for detection. In contrast to horizontal bounding boxes, a rotated bounding box includes an additional angle parameter $\theta$, as shown in Equation (2) compared to Equation (1). In the four-parameter representation, $Cx$ and $Cy$ represent the center coordinates of the bounding box, $h$ denotes the length of the bounding box's longer side, and $w$ represents the length of its shorter side. In the representation with five parameters, $Cx$ and $Cy$ denote the center coordinates of the bounding box, $h$ represents the length of the bounding box's longer side, $w$ represents the length of its shorter side, and $\theta$ represents the rotation angle of the rotated box, where $\theta$ is constrained within the range $\theta \in [-(\pi/2), (\pi/2))$. This paper adopts the convention of representing the longer side in OpenCV. The rotation angle, denoted as $\theta$, is the counterclockwise rotation angle relative to the horizontal axis (x-axis), representing the angle between the longer side of a rectangle (typically the wider side) and the horizontal axis. The coordinate system's origin is usually at the top-left corner, with the x-axis extending horizontally to the right and the y-axis vertically downward. In this system, counterclockwise rotation is indicated by negative angles, while clockwise rotation is represented by positive angles. Here, the longer side refers to the side of the rectangle with a length strictly greater than its adjacent side.

$$(Cx, Cy, w, h) \tag{1}$$

$$(Cx, Cy, w, h, \theta) \tag{2}$$

Hence, the YOLOv7's network architecture is not well-suited to rotated bounding box detection tasks, necessitating modifications to its network structure. The adjustments primarily involve two aspects: 1. Altering the model's parameter output from its original four parameters to five parameters. 2. Adapting the Intersection over Union (IOU) metric to accommodate rotated detection using poly-IOU, and subsequently modifying the corresponding code. Figure 2 illustrates the network architecture in this paper, while Figure 3 details the architecture further. The CBS module, as shown in (Figure 3a), is composed of a convolutional layer, a batch normalization layer (BN), and a Silu layer. Here, 'k' represents the number of convolutional kernels, while 's' indicates the stride during the convolution process. As illustrated in (Figure 3b), the ELAN module represents an efficient network structure. By regulating the shortest and longest gradient paths, this module enables the network to learn more features and enhances its robustness. (Figure 3c) represents the REP module. The REP module is divided into two parts: training and inference. The training module consists of three branches, namely a $3 \times 3$ convolution for feature extraction, a $1 \times 1$ convolution for feature smoothing, and an identity branch for directly transmitting information. The inference module consists of a $3 \times 3$ convolution, reparameterized based on the training module. (Figure 3d) is the MP module, comprising two branches. One branch undergoes max-pooling followed by a $1 \times 1$ convolution, while the other branch, after a $1 \times 1$ convolution, proceeds through a $3 \times 3$ convolution for down-sampling. Finally, the results from both branches are added together. (Figure 3e) consists of two parts; the role of the SPP module is to enlarge the receptive field through max-pooling, enabling the algorithm to adapt to images with different resolutions. In the first branch, four different scales of max-pooling are applied to assist the model in better recognizing targets of different sizes. The CSPC module divides features into two parts, with one undergoing conventional processing and the other undergoing SPP structure processing. Finally, the two parts are merged to reduce computational load, thereby improving speed and accuracy.

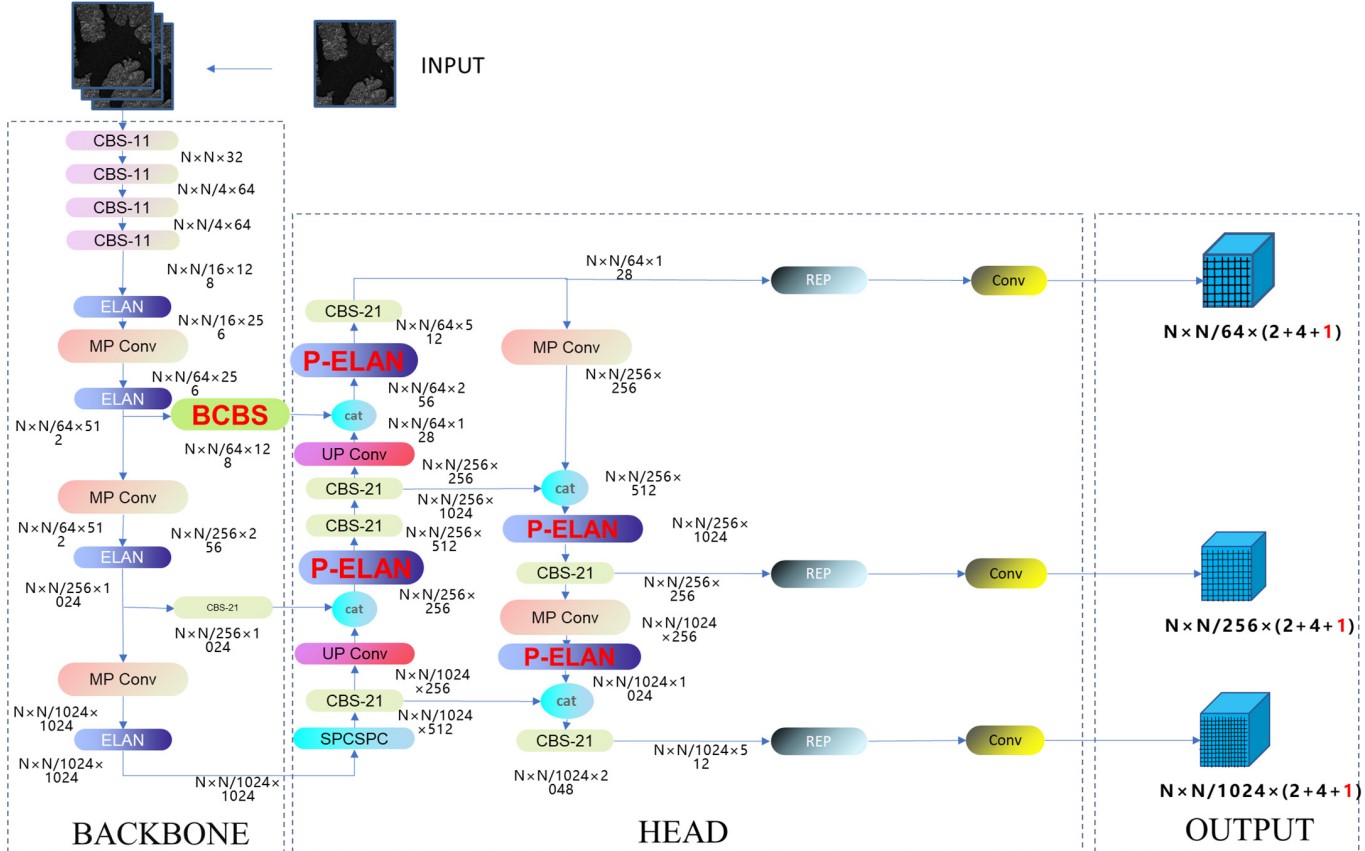

**Figure 2.** YOLOv7oSAR structure.

Due to the distinctive characteristics of rotated detection boxes, the conventional Intersection over Union (IOU) metric, which quantifies the spatial overlap between two bounding boxes by dividing the area of their intersection by the area of their union, may inadequately represent the overlap inherent in tilted or rotated bounding boxes. As a consequence, traditional IOU calculations may lack precision in evaluating the disparities between predicted and ground truth bounding boxes.

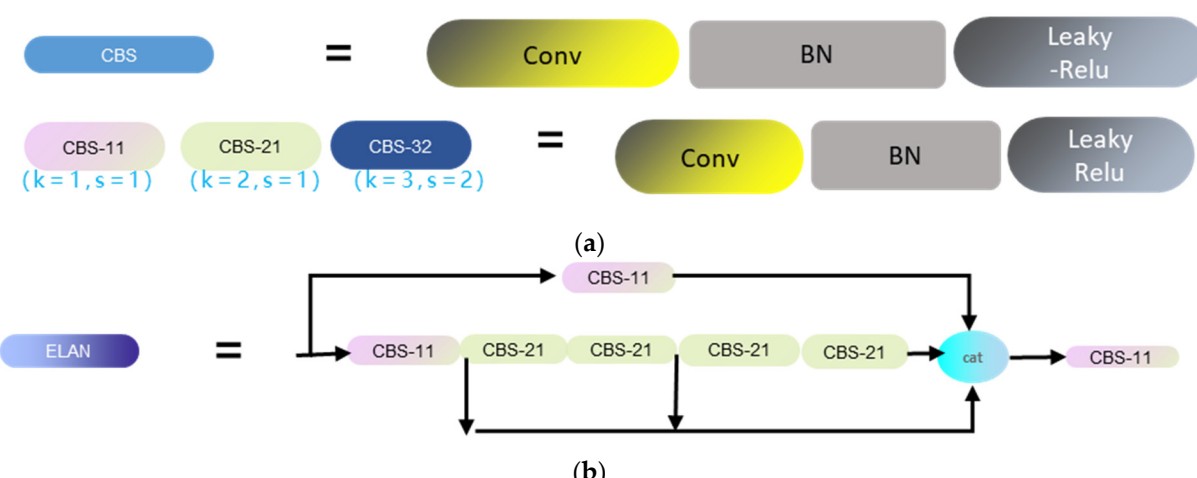

**Figure 3.** *Cont.*

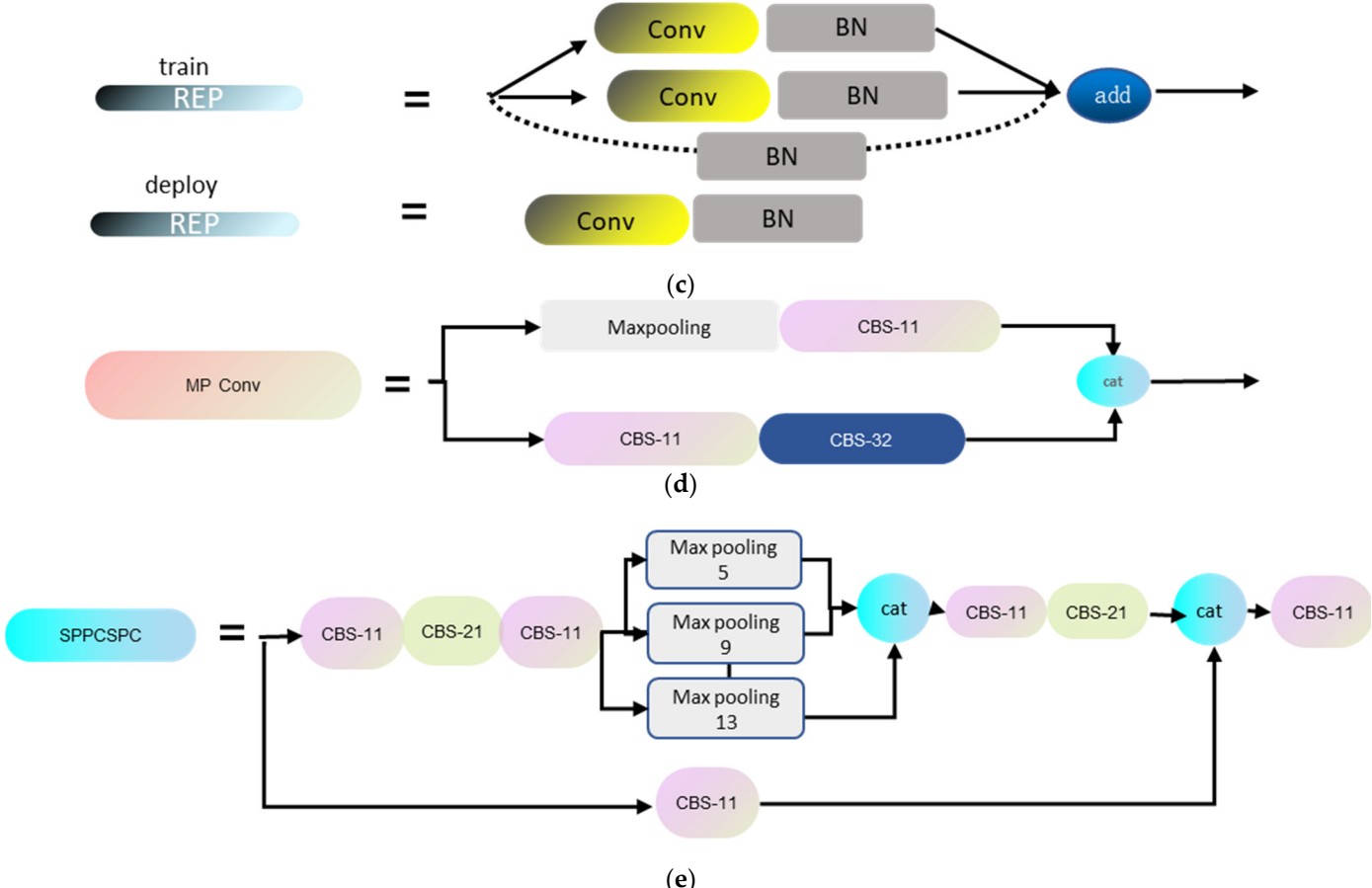

**Figure 3.** Structure of basic components of YOLOv7oSAR.

In response to this challenge, several scholars have introduced a range of rotated box detection algorithms, such as the Radar Region Proposal Network (RRPN) [39], one-stage anchorfree rotated object detector (FCOSR) [40], and ROI-Transformer [29], with the objective of approximating the nuanced distinctions between predicted and ground truth boxes. However, these approaches are subject to the limitations inherent in the traditional definition of rotated boxes, which encompass issues of angle and edge interchangeability, as well as class edge interchangeability. These constraints impede the accurate regression of predicted rotated bounding boxes.

To address these limitations, the Kullback–Leibler Divergence (KLD) loss function is employed, treating bounding boxes as Gaussian distributions and quantifying the relative entropy between the predicted and ground truth bounding boxes. This methodological refinement serves to overcome the challenges associated with conventional calculation methods, offering a more sophisticated and effective approach to the precise regression of rotated bounding boxes.

First, we need to convert the rotated bounding box $(Cx, Cx, w, h, \theta)$ into a 2D Gaussian distribution form $\mathcal{N}(m, \Sigma)$ (m represents the mean vector and $\Sigma$ represents the covariance matrix); R represents the rotation matrix and S represents the diagonal matrix of eigenvalues. Equations (3) and (4) are the detailed transformation formulas.

$$
\begin{aligned}
\Sigma^{1/2} &= RSR^{\top} \\
&= \begin{pmatrix} cos\theta & -sin\theta \\ sin\theta & cos\theta \end{pmatrix} \begin{pmatrix} \frac{w}{2} & 0 \\ 0 & \frac{h}{2} \end{pmatrix} \begin{pmatrix} cos\theta & sin\theta \\ -sin\theta & cos\theta \end{pmatrix} \\
&= \begin{pmatrix} \frac{w}{2}cos^2\theta + \frac{h}{2}sin^2\theta & \frac{w-h}{2}cos\theta sin\theta \\ \frac{w-h}{2}cos\theta sin\theta & \frac{w}{2}sin^2\theta + \frac{h}{2}cos^2\theta \end{pmatrix}
\end{aligned}
\tag{3}
$$

$$m = \left(x, y\right)^{\top} \tag{4}$$

Therefore, the Kullback–Leibler Divergence (KLD) loss is expressed as Equation (5):

$$
\begin{aligned}
D_{kl}\left(N_p || N_t\right) = \quad & \tfrac{1}{2}\left(\mu_p - \mu_t\right)^{\top} \Sigma_t^{-1} \left(\mu_p - \mu_t\right) \\
& + \tfrac{1}{2} Tr\left(\Sigma_t^{-1} \Sigma_p\right) + \tfrac{1}{2} ln \frac{|\Sigma_t|}{|\Sigma_p|} - 1
\end{aligned}
\tag{5}
$$

In this context, $N_p$ denotes the 2D Gaussian probability distribution associated with the predicted rotated bounding box, expressed as $X_p \sim \mathcal{N}_p(\mu_p, \Sigma_p)$. Here, $\mathcal{N}_p$ signifies the central position of the 2D Gaussian distribution, while $\mu_p$ represents the covariance matrix, $\Sigma_p$ providing insights into the distribution's shape, orientation, and correlation. Similarly, $\mathcal{N}_t$ represents the 2D Gaussian probability distribution linked to the true rotated bounding box, written as $X_t \sim \mathcal{N}_t(\mu_t, \Sigma_t)$. In summary, $\mu_t$ denotes the central position and $\Sigma_t$ denotes the covariance matrix of the 2D Gaussian distribution, adding details about the distribution's shape, orientation, and correlation.

### 2.3. Model Optimization

This section delineates optimization strategies with a dual focus: the augmentation of model precision and the mitigation of model resource consumption.

### 2.3.1. BCBS Module

Influenced by successful models in the vision Transformer category, such as the Vision Transformer (VIT) [41] and Detection Transformer (DETR) [42], remarkable achievements have been made in image processing tasks, demonstrating the potential of Transformers in the field of computer vision. However, traditional self-attention mechanisms face bottlenecks in terms of computation and memory complexity, making them challenging to apply effectively to relevant tasks. Inspired by the Swin-Transformer [43], BRA has emerged, featuring adaptive sparse queries. This innovation allows the model to efficiently query key regions and aggregate attention with a smaller parameter count and faster speed, thereby enhancing the model's capability in detecting small objects.

The formulas for traditional attention mechanisms are represented as Equations (6) and (7):

$$
\begin{aligned}
\text{Attention}(Q, K, V) &= \text{softmax}\left(\frac{QK^T}{\sqrt{C}}\right)V \\
\text{MHSA}(X) &= \text{Concat}(\text{head}_0, \text{head}_1, \dots, \text{head}_h)W^o
\end{aligned}
\tag{6}
$$

$$
\text{head}_i = \text{Attention}\left(XW_i^q, XW_i^k, XW_i^v\right) \tag{7}
$$

In the context of attention mechanisms, three key matrices are utilized: the query matrix $Q \in R^{N_q \times c}$, the key matrix $K \in R^{N_{kv} \times C}$, and the weight matrix $V \in R^{N_{kv} \times C}$. Here, $N_q$ represents the number of query vectors, $N_{kv}$ denotes the number of key vectors (and corresponding value vectors), and $C$ signifies the number of input channels. The softmax function is applied to each row during computation, and $\sqrt{C}$ is employed to maintain gradient stability during training.

In Transformer models, the multi-head attention mechanism, represented in Equation (7), involves the matrices Q, K, and V, derived from a common input tensor $X \in \mathbb{R}^{N \times C}$. Here, $N = H \times W$, where $H$ and $W$ represent the height and width of the spatially unfolded feature map. The tensor is divided into h segments along the channel dimension, forming "heads" denoted as $\text{head}_i \in \mathbb{R}^{N \times \frac{C}{h}}$. The projection weights $W_i^q$, $W_i^k$, and $W_i^v \in \mathbb{R}^{C \times \frac{C}{h}}$ shape the attention mechanism. A linear transformation, $W^o \in \mathbb{R}^{C \times C}$, combines the outputs across heads. The computational complexity is $O(N^2)$.

To address this issue, BRA adopts a dynamic, query-aware sparse attention mechanism. This attention mechanism filters out irrelevant key–value pairs at a coarse-grained level, retaining a small set of key routing regions. Fine-grained one-to-one attention calculations

are then performed on the union of these routing regions. The implementation of attention is as follows: For a given feature map $X^r \in \mathbb{R}^{S^2 \times \frac{HW}{S^2} \times C}$, we partition the feature map X into non-overlapping regions of size $S \times S$ by reshaping it into $Q, K, V \in \mathbb{R}^{S^2 \times \frac{HW}{S^2} \times C}$. We obtain tensors for the queries, keys, and values of $X^r$ as shown in Equations (8)–(10):

$$Q = X^r W^q \tag{8}$$

$$K = X^r W^k \tag{9}$$

$$V = X^r W^v \tag{10}$$

where $W^q$, $W^k$, and $W^v \in \mathbb{R}^{C \times C}$ are the weight matrices for Q, K, and V, respectively.

Subsequently, a regional aggregation process is applied by computing the average values for both Q and K, Q, and K yielding region-level queries and key values denoted as $Q^r, K^r \in \mathbb{R}^{S^2 \times C}$. Subsequently, the matrix product of $Q^r$ and the transposed $K^r$ result in the formation of the region adjacency matrix $A^r$. In this matrix, the individual elements signify the degree of association between respective regions.

Following this, a selective region pruning strategy is implemented, wherein only the top k connections for each region are retained. This process culminates in the derivation of the routing index matrix $I_r \in \mathbb{N}^{S^2 \times k}$, where $S^2$ denotes the total number of partitioned regions and k signifies the quantity of preserved connections chosen for retention.

Regarding the obtained routing index matrix $I_r$, we conduct fine-grained one-to-one attention calculations. This operation focuses on the top K key regions for each region, aggregating their corresponding key values. The formulas are as shown in Equations (11) and (12):

$$K^g = gather(K, I^r) \tag{11}$$

$$V^g = gather(V, I^r) \tag{12}$$

For $K^g, V^g \in \mathbb{R}^{S^2 \times \frac{kHW}{S^2} \times C}$, $K^g$ and $V^g$ represent the aggregated tensors of the keys and values and the subsequent attention operation is defined as shown in Equation (13):

$$O = Attention(Q, K^g, V^g) + LCE(V) \tag{13}$$

In this context, O denotes the resultant output. The designation "Local Context Enhancement" (LCE) characterizes the introduced operation dedicated to augmenting the local context. Within this attention mechanism, the convolutional kernel size is explicitly set to 5. Figure 4 illustrates the visual effect of the Boundary Refinement Attention (BRA) mechanism, which progressively filters the entire image from coarse to fine, focusing on small target areas. Figure 5 shows the BCBS structure, where the BRA (Boundary Refinement Attention) mechanism is embedded into the CBS (Contextual Block and Spatial Attention) structure. This structure allows for on-the-fly integration and can be easily embedded into any other structure.

### 2.3.2. Model Scaling and the P-ELAN Module

Typically, a reduction in model size involves strategies such as parameter pruning and knowledge distillation, aiming to mitigate computational demands during model deployment. However, these methodologies often entail complex procedures, necessitating extensive experimental tuning, and are susceptible to substantial performance variations based on the dataset. Consequently, this study proposes a compression strategy tailored to the current model. The overarching goal is to diminish the model parameters without compromising accuracy, thereby effectively reducing the inference time of the model.

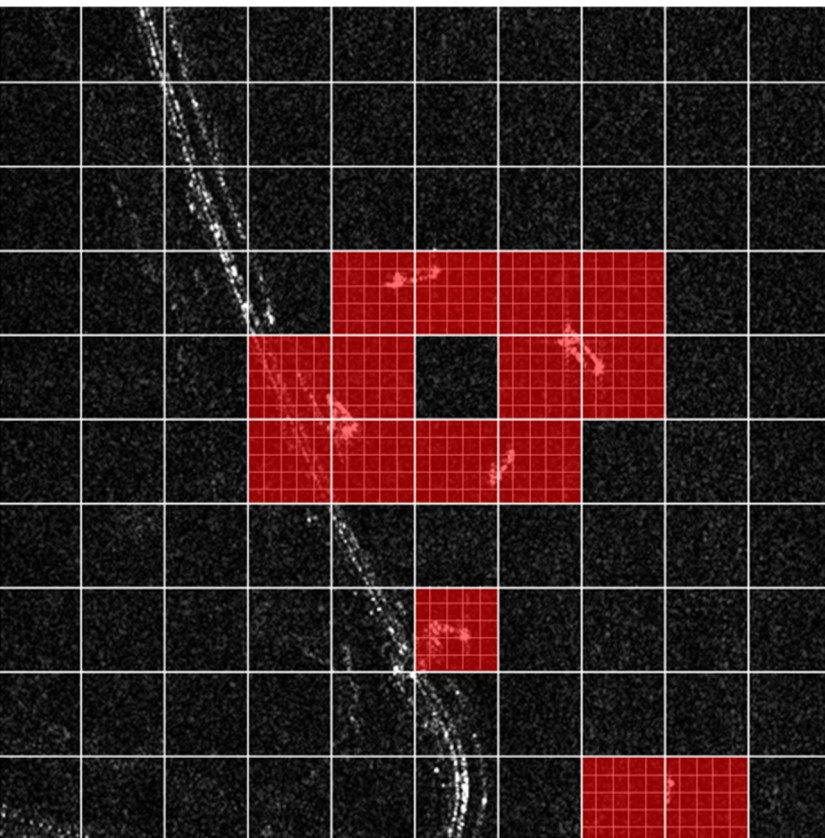

**Figure 4.** At a coarse-grained level, this mechanism purposefully filters out low-relevance key-value pairs, selectively retaining crucial routing regions. Following this initial filtration, detailed one-to-one attention calculations are specifically conducted within these identified areas. The depicted strategic attention mechanism in the figure emphasizes the model's focus on ship regions (highlighted in red) and performs fine-grained search query operations while effectively disregarding background elements (depicted in black).

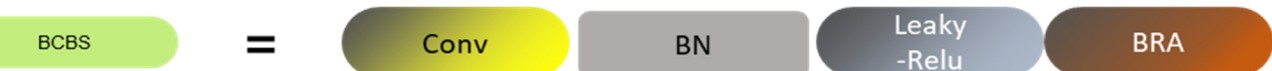

**Figure 5.** BCBS structure.

Model scaling represents a significant paradigm for resizing models effectively. This approach involves the fine-tuning of model parameters such as width, depth, resolution, and the count of feature pyramids to achieve a nuanced balance between network parameters, computational overhead, inference speed, and accuracy. Notably, YOLOv7 incorporates a substantial number of concatenation structures, prompting the need for a redefined approach to model scaling in its application. In this study, we adhere to the original YOLOv7 network architecture and leverage the scaling factor configurations from YOLOv7-tiny for network training.

To achieve fast neural networks with low latency and high throughput, many researchers and practitioners have focused on optimizing their convolutional structures. Notable examples include MobileNet [44], ShuffleNet [45], and GhostNet [46], which are based on improved convolutional architectures such as Depthwise Separable Convolution (DWConv)and Group Convolution (GConv). While these methods effectively reduce the floating-point computation and inference time of models, an excessive emphasis on minimizing floating-point computations introduces a plethora of additional operations, including concatenation, shuffling, and pooling. Ironically, this has led to the observation that, in practical applications, the inference speed of models has not decreased but, in fact, the

increased latency has contributed to longer model inference times. To ensure that reducing parameters does not affect the model's speed, this section introduces PConv, a convolutional structure designed to reduce redundant computations and optimize model memory access. The DWConv is a common variant in convolutional structures that separates convolutional kernel operations. This design aims to reduce floating-point computations by dividing the convolution into depthwise and pointwise convolutions. As a result, the total number of floating-point operations in the computational device is significantly reduced compared to conventional convolutions.

For example, considering an input $I \in \mathbb{R}^{c \times h \times w}$, where c denotes the number of channels, h represents the image height, and w signifies the image width, the floating-point operations for Depthwise Separable Convolution are expressed as $h \times w \times k^2 \times c$. In contrast, regular convolution necessitates $h \times w \times k^2 \times c$. Despite the evident decrease in computational workload, precision maintenance mandates an expansion in the convolution's channel count. This augmentation, unfortunately, results in heightened memory access, subsequently impeding the model's speed. This predicament is especially consequential for edge devices.

To address this issue, this paper introduces the PConv structure, which simultaneously reduces computational redundancy and memory access. It applies regular convolution to only a subset of input channels for spatial feature extraction while keeping the remaining channels unchanged. For consecutive or regular memory access, this convolution considers the first or last consecutive $c_p$ channels as representatives of the entire feature map for computation. Assuming the input and output feature maps have the same number of channels, the floating-point operations (FLOPs) of PConv are only related to the number of channels in the feature map.

$$h \times w \times k^2 \times c_p^2 \tag{14}$$

For a typical partial ratio $r = \frac{c_p}{c} = \frac{1}{4}$, PConv's floating-point operations (FLOPs) are only $\frac{1}{16}$ of regular Conv. Additionally, PConv has a smaller memory footprint, i.e., $h \times w \times 2c_p + k^2 \times c_p^2 \approx h \times w \times 2c_p$; its memory access is only $\frac{1}{4}$ that of regular Conv. Figure 6 demonstrates the transformation of the CBS (Contextual Block and Spatial Attention) structure by replacing the Conv structure with the Pconv structure. Additionally, by replacing the CBS structure in the ELAN (Enhanced Local-Attention Network) structure, a P-ELAN structure is obtained.

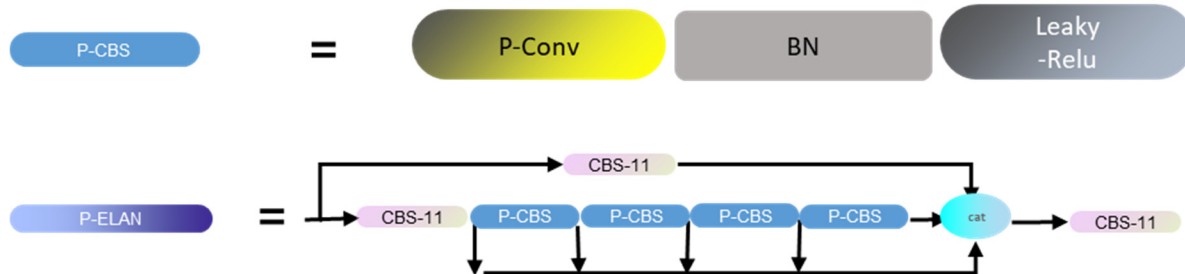

**Figure 6.** P-ELAN structure.

## 3. Experimental Design

### 3.1. Experimental Setup

All experiments in this study were conducted on the Ubuntu operating system with the following specific configuration:

- Operating System: Ubuntu 20.04 LTS—we chose Ubuntu 20.04 LTS for its stability and widespread support in the community.
- Processor: Intel (R) Core (TM) i9-10900X CPU @ 3.70GHz—our hardware configuration includes a high-performance Intel Xeon processor, providing reliable computational capabilities.

- Memory: 128 GB—additionally, we equipped the system with 128 GB of memory to ensure no memory constraints during large-scale deep learning model training.
- Graphics Card: NVIDIA GeForce RTX 3090 with 24 GB of VRAM—this graphics card offers excellent computational performance and parallel processing capabilities, significantly improving training efficiency.

Regarding software tools, we utilized PyTorch 1.7 for model construction and training. PyTorch is a popular deep learning framework known for its rich features and user-friendly API, facilitating convenient model construction and training processes. To accelerate computations, we employed CUDA 11.0 for GPU acceleration, fully leveraging the parallel computing capabilities of the RTX 3090 graphics card and significantly reducing model training time.

### 3.2. Dataset

To authentically evaluate the model's performance, this study utilized the publicly available Synthetic aperture radar Ship Detection Dataset (RSDD) for training and validation. This dataset encompasses a rich variety of data sources, including diverse imaging modalities and polarization techniques, along with a well-distributed set of ship instances. Beyond the conventional perspective of vessels in open waters, this dataset covers various scenarios, including coastal regions, ports, and islands. Due to the limited availability of publicly accessible data for ship detection with slant-range annotations [23,47–50], a proprietary dataset was created to further assess the model's generalization capabilities, focusing on ship classification and detection.

### 3.2.1. RSDD—The Rotated Ship Detection Dataset in SAR Images

The RSDD, meticulously curated and publicly disclosed by the research team led by Professor Xu Cong'an at the Naval Aviation University, encompasses a comprehensive compilation of 127 scenes. Among these, 84 scenes are derived from the high-resolution Gaofen-3 satellite, while an additional 41 scenes originate from TerraSAR-X, complemented by 2 uncropped large-scale images. This dataset is characterized by its richness in terms of diverse imaging modalities, polarization modes, and resolutions, featuring a total of 7000 meticulously sliced images. This dataset can be downloaded at https://radars.ac.cn/web/data/getData?dataType=SDD-SAR accessed on 27 February 2024.

A distinctive feature of the RSDD lies in its contemporaneity, reflecting recent ship instances with arbitrary rotation angles, substantial aspect ratios, a notable prevalence of diminutive targets, and a wealth of diverse contextual scenes. The annotation process, a fusion of automated procedures and meticulous manual correction, attests to the dataset's efficiency in providing accurate and detailed ship instance annotations. This dataset serves as an invaluable resource for advancing the research in SAR ship detection, offering a faithful representation of real-world data complexities.

### 3.2.2. SAR Image Ship Verification Dataset

In order to verify the robustness of our model, this paper created a simplified dataset following the approach of the RSDD model. The following is an introduction to this dataset: Pearl Harbor is located approximately 15 miles southwest of Honolulu, Hawaii, and is one of the most renowned harbors in the United States. Serving as Hawaii's largest port, the geographical coordinates of Pearl Harbor range between approximately 21.3°N and 21.5°N latitude and 157.9°E and 157.95°E longitude. The harbor's strategic importance and rich historical background have established it as a crucial military base in Hawaii. Throughout history, Pearl Harbor has housed various types of warships, including battleships, aircraft carriers, destroyers, submarines, and more. Currently, the primary types of ships docking at Pearl Harbor include destroyers, submarines, support ships, and others. This dataset aims to conduct a simple evaluation of the detection and classification capabilities of our target detection model on another dataset. Figure 7 illustrates the geographical location of the dataset selection area and a comparison of various types of vessels under SAR and in

optical images. In this study, a simple classification of different types of vessels under SAR images was conducted by consulting the literature and using optical images for assistance.

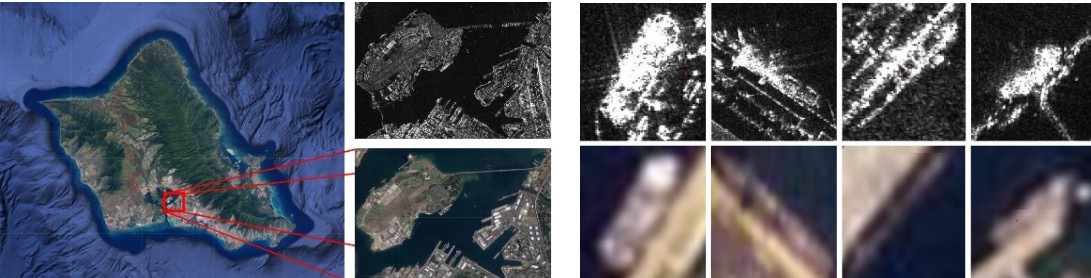

(**a**) Study Area Overview    (**b**) Ship Classification Data Comparison

**Figure 7.** The upper-left image (**a**) displays the actual study area, the upper-right image shows high-resolution data from the Gaofen-3 satellite, and the lower-right image shows data from the Planet optical satellite. In (**b**), a comparison of various types of vessels under SAR and in optical images is presented. The upper section illustrates different types of vessels under the Gaofen-3 satellite, while the lower section showcases various types of vessels under the Planet optical satellite. The categories, from left to right, are a destroyer, support ship, submarine, and ship (encompassing civilian and research vessels).

The data preprocessing pipeline comprises five key steps: the complex data conversion of SAR images, multi-view processing, filtering and denoising, orthorectification, and channel expansion. Unlike traditional optical images, SAR images typically have only one channel. Since model training requires reading image information through OpenCV, which can only read conventional RGB images, for the convenience of model training the final preprocessing step involves duplicating the original single-channel image twice to expand it into a three-channel image. Given the often substantial size of remote sensing images, direct detection from the original image is infeasible. Consequently, a slicing process precedes detection, wherein an overlap ratio, typically ranging from 40% to 70%, is set to mitigate potential target omission.

Popular annotation software such as Labelme v5.4.1 and roLabelimg v3.0 can meet these requirements. Labelme is suitable for annotating horizontal bounding boxes, while ro-labelimg is effective for annotating rotated bounding boxes. In the case of smaller datasets, manual annotation using software is feasible. For more extensive datasets, it is recommended to use the Polar Encodings SAR ship rotated bounding box detection model for rough annotation. Table 1 displays the detailed data types used in the dataset, ultimately creating a small SAR image dataset with rotated bounding boxes for ship classification, consisting of 2000 slices.

**Table 1.** Validation dataset's data format.

| Sensor | Longitude | Latitude | Time | Imaging Mode | Resolution (m) | Polarization Mode | Product Level | Incidence Angle (°) | Imaging Width (km) |
|---|---|---|---|---|---|---|---|---|---|
| GF-3 | W158.0 | N21.4 | 20220228 | SL | 1 | HH | L2 | 20–50 | 10 |
| GF-3 | W158.0 | N21.4 | 20220313 | SL | 1 | HH | L2 | 20–50 | 10 |
| GF-3 | W158.0 | N21.4 | 20220415 | SL | 1 | HH | L2 | 20–50 | 10 |
| GF-3 | W158.0 | N21.4 | 20220515 | SL | 1 | HH | L2 | 20–50 | 10 |
| GF-3 | W157.9 | N21.4 | 20220613 | SL | 1 | HH | L2 | 20–50 | 10 |

### 3.3. Evaluation Criteria

The dataset is divided into three parts: the training set, validation set, and test set. The division ratio can be adjusted appropriately based on the dataset's size. In this experiment,

the ratio is set as 7:2:1. Hyperparameters are the parameters set before the training process begins, and they can be adjusted based on the training results. After adjustments, the model is retrained. The number of training epochs in this study is set to 300, the batch size is 4, and the image size is $512 \times 512$. Other hyperparameters are fine-tuned based on the YOLOv7's official model hyperparameters.

The final result is evaluated based on the detection evaluation metrics of the COCO dataset, which include average precision (AP), $AP_{50}$, $AP_{75}$, APS, APM, and APL. $AP_{50}$ represents the average precision at an IOU (Intersection over Union) threshold of 0.5, while $AP_{75}$ represents the average precision at an IOU threshold of 0.75. AP represents the average precision at different IOU thresholds (0.5, 0.55, 0.60, 0.65, 0.70, 0.75, 0.80, 0.85, 0.90), and, in this study, the $AP_{50}$ is chosen as the detection accuracy metric.

In this detection task, the IOU is replaced by the KLD IOU to measure the overlap between a detection box and a ground truth box. For a given KLD IOU threshold, the average precision (AP) is the mean of the precision values across the precision–recall (P–R) curve. Here, P denotes precision and R denotes recall. The specific expressions are as shown in Equations (15)–(18):

$$precision = \frac{TP}{TP + FP} \tag{15}$$

$$recall = \frac{TP}{TP + FN} \tag{16}$$

$$AP = \int_{0}^{1} P(r)dr \tag{17}$$

$$mAP = \frac{1}{N}\sum_{i=1}^{N} AP(i) \tag{18}$$

where TP denotes the count of positive instances within the detection box, FP signifies the count of false positives (also referred to as negative instances) within the detection box, and FN represents the count of missed detections. The sum of TP and FN equals the total count of the ground truth boxes. $P(r)$ represents the precision–recall curve, consisting of multiple sets of precision and recall. It is crucial to note that the P–R curve is often discontinuous in practical computations. Therefore, AP typically denotes the average of all precision values along the P–R curve.

To assess the model's parameter count and computational requirements, this study calculates the model's parameter count and floating-point operations (FLOPs). The model parameter count refers to the total number of trainable parameters in a deep learning model, including weights and biases, and is commonly used to measure the model's parameterization. FLOPs represent the number of floating-point operations performed per second. In deep learning, this is often used to gauge the model's computational complexity, indicating the number of floating-point operations executed during the inference or training process. The quantity of FLOPs is typically closely related to the model's computational demands and speed. A higher number of FLOPs generally implies a larger computational burden, necessitating better hardware.

The formulas for calculating the parameter count for each convolutional layer are as shown in Equations (19) and (20):

$$\text{Convolutional Layer params} = (C_o \times (k_w \times k_h \times C_i + 1)) \tag{19}$$

$$\text{Fully Connected Layer params} = (C_0 \times (C_i + 1)) \tag{20}$$

where $C_o$ represents the number of output channels, $C_i$ represents the number of input channels, $k_w$ represents the width of the convolutional kernel, and $k_h$ represents the height

of the convolutional kernel. The term $w \times h \times C_i$ inside the parentheses denotes the number of weights in a single convolutional kernel. The parameter count for the bias is represented by adding one to the count of a single convolutional kernel's parameters, and thus the parentheses encapsulate the parameter count for a single convolutional kernel. When performing batch normalization, bias is not needed. In such cases, there is no need to add one to the count in the formula. In fully connected layers, the parameter count is directly obtained by calculating the product of the output channel number and the input channel number, plus one.

The FLOPs calculations for each layer of the convolutional and fully connected layers are as shown in Equations (21) and (22):

$$\text{FLOPs for Convolutional Layer} = [(C_i \times k_w \times k_h) + (C_i \times k_w \times k_h - 1) + 1] \times C_o \times W \times H \tag{21}$$

$$\text{FLOPs for Fully Connected Layer} = (2C_i - 1) \times C_m \tag{22}$$

where the value of $[(C_i \times k_w \times k_h) + (C_i \times k_w \times k_h - 1) + 1]$ represents the computational cost (multiplications and additions) required to calculate one point in the feature map through a convolutional operation. $C_i \times k_w \times k_h$ denotes the multiplication operations in a single convolutional kernel, and $C_i \times k_w \times k_h - 1$ represents the addition operations in a single convolutional kernel. The parameter count for the bias is represented by adding one to the count of a single convolutional kernel's parameters. $W$ and $H$ are the length and width of the feature map, and $C_o \times W \times H$ represents the total number of elements in the feature map. In a fully connected layer, the computational cost is directly calculated by multiplying two times the output channel number by the input channel number minus one.

The computational complexity of the model in this paper primarily consists of the sum of 101 convolutional layers and 3 fully connected layers. As each convolutional layer has different parameters, it is necessary to calculate the complexity for each layer separately. However, detailed calculations for each layer are not provided here. An approximate estimation can be obtained through Equations (21) and (22), and the calculation formula is as follows (23):

$$\text{Model Computational Complexity} = \sum_{i=1}^{101} \left( [(C_i \times k_w \times k_h) + (C_i \times k_w \times k_h - 1) + 1] \right) + \sum_{i=1}^{3} \left( (2C_i - 1) \times C_m \right) \tag{23}$$

## 4. Results

### 4.1. Ablation Experiment

This study performed model scaling by adjusting both width and depth, with the depth multiple set to 0.30 and the width multiple set to 0.5. Compared to the original model parameters, where all factors were set to 1, this configuration achieved comparable accuracy. The rationale behind these adjustments lies in the fact that the original model was designed for optical image detection, whereas SAR images lack the visual richness found in their optical counterparts. Thus, an oversized model is unnecessary for detecting ships in SAR images. Therefore, the chosen scaling factors in this study are deemed judicious, reducing the original model's parameter size while maintaining an equivalent accuracy to the model set with the original parameters. All enhancements in this work are grounded in these scaling factors' modifications to the model's coefficients.

To further enhance the ship detection capabilities of the model, we introduced the BRA mechanism and embedded it within CBS, proposing the BCBS structure. The BRA mechanism allows the model to rapidly focus on key image areas, improving its proficiency in detecting small targets, with only a marginal increase in the model's parameter size. This study incorporates it into the shallow feature layers of the model to enhance its small target detection capability. Additionally, for model lightweighting, we not only adjusted the model's structure by reducing its width and depth but also introduced the PConv structure. We refined the Efficient Layer Aggregation Networks (ELAN) module into the

PConv ELAN (P-ELAN) module. In experiments, by replacing the ELAN in the head region with P-ELAN, we achieved a reduction in model parameters and greater computational efficiency. To validate the effectiveness of these improvements, this study conducted ablation experiments, and the results are presented in Table 2 and Figure 8:

**Table 2.** Performance metrics of YOLOv7oSAR variants.

| Module/Metric | $AP_{50}$ Offshore | $AP_{50}$ Nearshore | Param (M) | FLOPs |
|:---:|:---:|:---:|:---:|:---:|
| **YOLOv7oSAR** | 0.927 | 0.586 | 9.14 M | 16.6 GFLOPs |
| **YOLOv7oSAR + BRA** | 0.949 | 0.660 | 9.16 M | 16.7 GFLOPs |
| **YOLOv7oSAR + PConv** | 0.920 | 0.534 | 8.21 M | 15.4 GFLOPs |
| **YOLOv7oSAR + BRA + PConv** | 0.938 | 0.634 | 8.23 M | 15.5 GFLOPs |

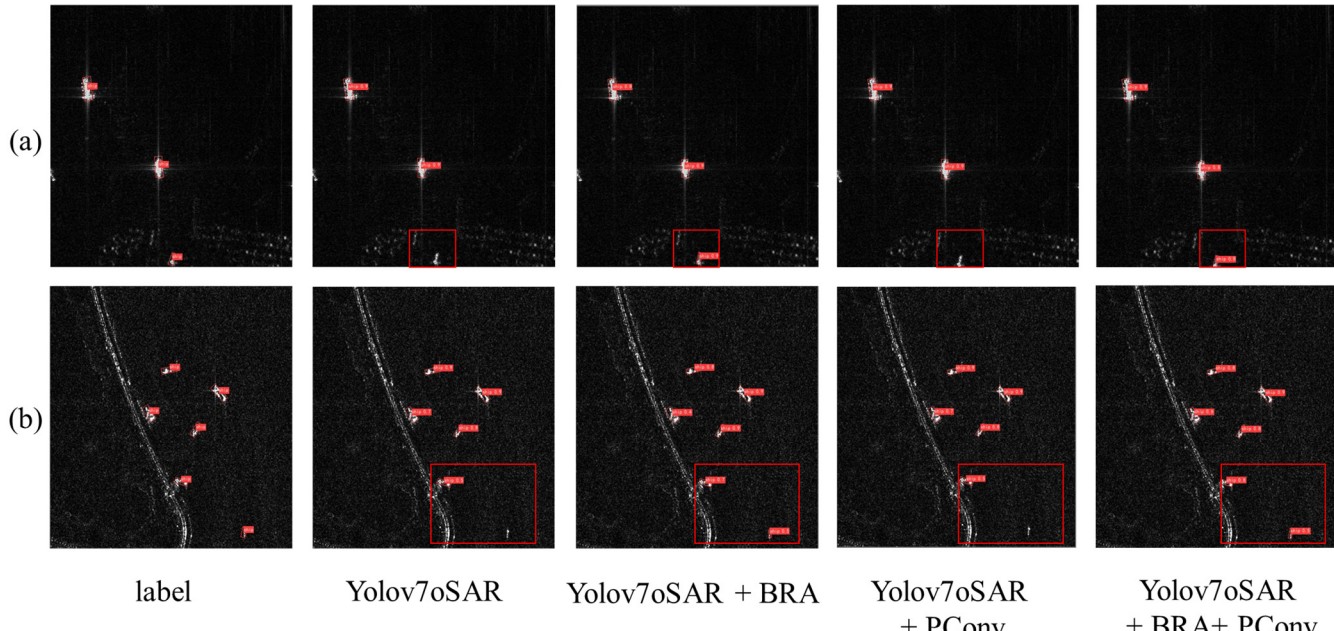

|  |  |  |  |  |
|:---:|:---:|:---:|:---:|:---:|
| label | Yolov7oSAR | Yolov7oSAR + BRA | Yolov7oSAR + PConv | Yolov7oSAR + BRA+ PConv |

**Figure 8.** Ablation experiment: (**a**) represents the offshore area, while (**b**) represents the nearshore area. The red boxes indicate regions prone to missed detections.

The model's accuracy in offshore detection improved by 2.2%, and its nearshore detection accuracy increased by 7.4% after incorporating the BRA attention mechanism. Table 1 indicates that the improvement in accuracy is higher for the nearshore area than for the offshore area. This is likely due to the more complex background in nearshore areas compared to offshore ones, making the BRA attention mechanism more pronounced and resulting in a significant enhancement in nearshore detection. From Figure 8, it is evident that the model with the BRA attention mechanism excels in detecting small vessels. Small vessels that were initially undetectable can now be successfully identified, whether nearshore or offshore.

After introducing the PConv convolutional structure, the model's parameter count decreased by 7.8%, and its computational load decreased by 10.1%. However, there was a 2.7% decline in offshore accuracy and 5.2% decrease in nearshore accuracy. This decline may be attributed to the complexity of nearshore scenes, where the model, after reducing its parameters, experienced a slight decrease in its ability to accurately locate objects in overly intricate scenarios. When incorporating all modules, including the BRA and PConv, into the model, its accuracy increased by 1.1% for nearshore and decreased by 5% for offshore detection. Its parameter count decreased by 10.1%, and its computational load decreased by 7.1%. The PConv module can be replaced based on the specific requirements of practical

applications. In this study, only the ELAN structure in the model's head network was replaced with the P-ELAN structure.

*4.2. Comparative Experiment*

Our proposed method has been compared with several advanced models. For instance, the Gliding Vertex model accurately describes multi-oriented objects by sliding the vertices on the horizontal bounding box, avoiding the confusion caused by direct regression. It introduces length ratio regression and an obliquity factor based on the area ratio, enhancing the model's robustness to rotated targets [30]. Redet presents a rotation-equivariant detector, achieving precise orientation prediction and significantly reducing model size. It introduces a rotation-invariant RoI Align for feature extraction, adapting to the RoI's direction [27]. S2ANet incorporates a rotation-equivariant network for accurate arbitrary-oriented object detection. It contains a rotation-invariant RoI Align module for adaptive feature extraction based on the RoI's direction [28]. ROI-Transformer introduces a single-stage alignment network with feature alignment and oriented detection modules. It refines its anchors for high-quality anchor generation and aligns convolutional features adaptively with a novel alignment convolution [29]. Rotated FCOS includes a single-stage alignment network, improving its detection accuracy for rotated objects. It uses active rotating filters for encoding directional information, producing orientation-sensitive and invariant features [40]. Rotated RetinaNet extends the RetinaNet framework, efficiently detecting rotated objects with rotated anchors and a skewed IOU. It introduces a refinement network and feature alignment module, enhancing detection accuracy [20]. The Rotated Faster R-CNN combines a horizontal bounding box prediction and a dedicated regression branch for oriented bounding box. This achieves more accurate and comprehensive detection outcomes through multitask learning [51]. These models demonstrate unique technical designs, each excelling in handling challenging arbitrarily oriented object detection tasks. These models' comparison results and visual effects are shown in Table 3 and Figure 9.

**Table 3.** Comparison of experimental results.

| Module/Metric | $AP_{50}$ Offshore | $AP_{50}$ Nearshore | Param (M) | FLOPs |
|---|---|---|---|---|
| YOLOv7oSAR + BRA | 0.949 | 0.660 | 9.16 M | 16.60 GFLOPs |
| Gliding vertex | 0.810 | 0.640 | 41.13 M | 63.25 GFLOPs |
| Redet | 0.900 | 0.590 | 31.64 M | 40.98 GFLOPs |
| S2anet | 0.790 | 0.450 | 38.54 M | 63.25 GFLOPs |
| ROI-Transformer | 0.897 | 0.456 | 55.03 M | 77.15 GFLOPs |
| Rotated FCOS | 0.900 | 0.635 | 31.89 M | 51.55 GFLOPs |
| Rotated RetinaNet | 0.898 | 0.638 | 36.42 M | 53.98 GFLOPs |
| Rotated Faster R-CNN | 0.900 | 0.640 | 41.14 M | 63.26 GFLOPs |

The results presented in Table 3 underscore the performance of our research model. Achieving an offshore detection accuracy of 94.9%, it surpasses the lowest-performing model, S2ANet, by a significant margin of 15.9% and outperforms the highest-performing offshore model, Redet, by 4.9%, with its detection accuracy of 90%. For nearshore detection accuracy, our model attains 68.4%, exhibiting a notable superiority over the least performing nearshore model, S2ANet, by 23.4%, and demonstrating a modest 4.4% advantage over the highest-performing offshore detection model, Rotated FCOS, which has a detection accuracy of 90%. Considering the models' parameterization, our proposed model exhibits a parameter count that is one-third that of the minimal parameter model, Redet, and one sixth of the maximal parameter model, ROI-Transformer. Regarding computational complexity, our model has one third of the computational load of the least complex model, Redet, and one fifth of the computational load of the most complex model, ROI-Transformer. These findings underscore the efficiency and effectiveness of our model in achieving a competitive performance with reduced computational demands.

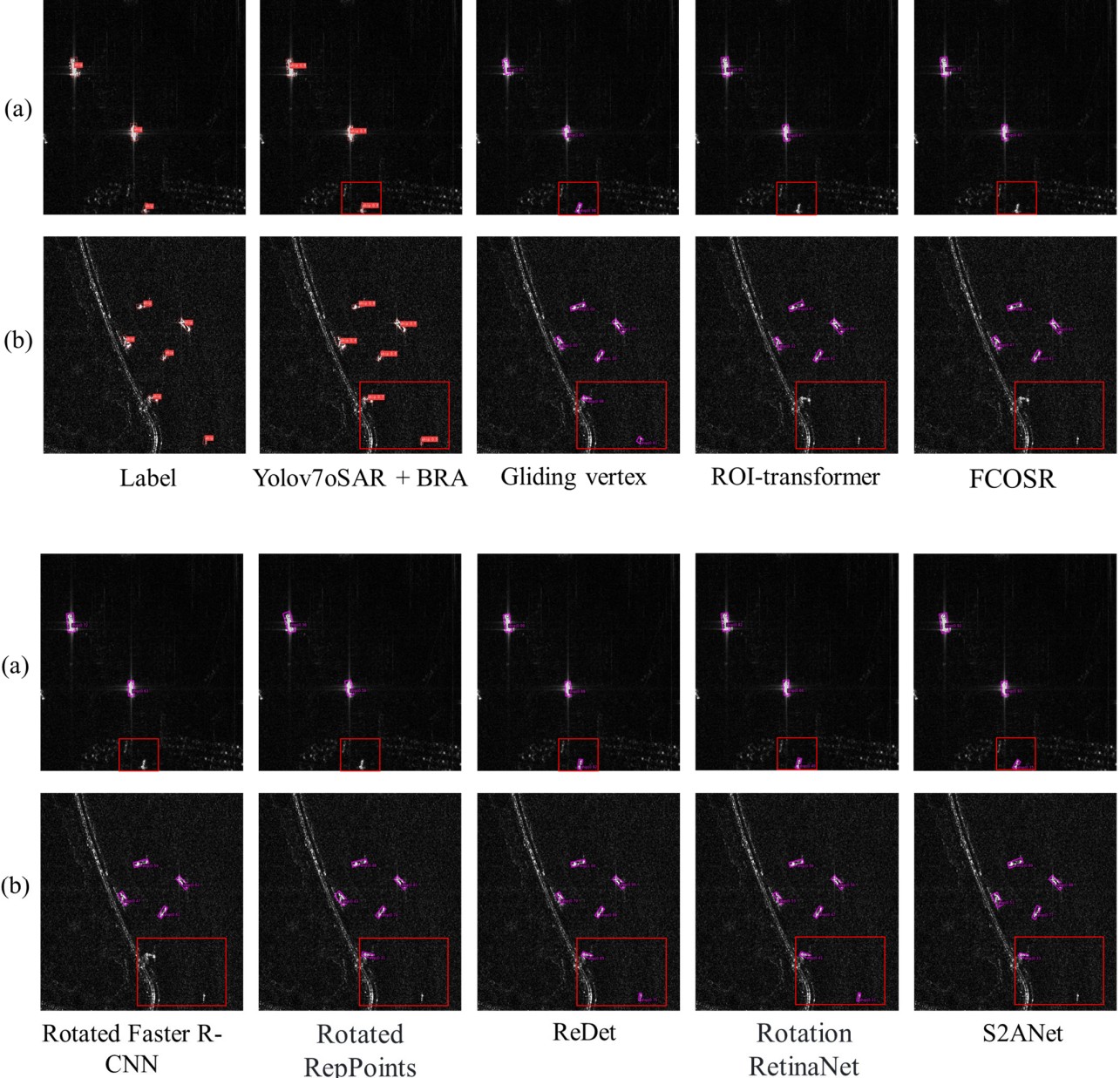

**Figure 9.** Comparative experiment: (**a**) represents the offshore area, while (**b**) represents the nearshore area. The red boxes indicate regions prone to missed detections.

In observing Figure 9, it is evident that the Gliding Vertex model demonstrates a comprehensive and accurate detection of all ships without any instances of missed detections. However, during the detection of offshore ships, there is a notable occurrence of square-like bounding boxes for ships located in the middle position. Similarly, the bottommost ship in nearshore detection exhibits a square-like bounding box. Contrastingly, models such as ROI-Transformer, Rotated FCOS, and Rotated Faster R-CNN exhibit a more pronounced occurrence of missed detections. Specifically, in the offshore ship detection scenario, ships positioned at the bottom are not accurately identified. In nearshore ship detection, both the ships on the bottom left and the bottommost ship are not precisely detected. The Rotated RepPoints model shows fewer instances of missed detections. In offshore ship detection, ships positioned at the bottom are not accurately detected, and, in nearshore ship detection, the bottommost ship is also not precisely identified. Models like Redet and Rotated RetinaNet exhibit an overall absence of missed detections, showcasing a robust detection

performance. However, these models have significantly higher parameter counts and computational loads compared to the model proposed in this study. According to the data in Table 3, their detection accuracy also falls short of the precision achieved by our research model. S2ANet, while displaying commendable nearshore detection, exhibits a suboptimal performance in offshore detection. In summary, while most models exhibit occasional instances of missed detections, our proposed model demonstrates a superior performance in terms of both minimizing missed detections and overall detection effectiveness.

### 4.3. Verification Experiment

To evaluate the detection and classification performance of our experimental model in real-world scenarios, this study conducted performance tests on a self-made dataset. The model's results on the self-made dataset, as shown in Figure 10 and Table 4, demonstrated an overall detection accuracy of 94.2%. The accuracy for each ship category is as follows: destroyer—73.7%, support ship—94.4%, ship (civilian, research, etc.)—100.0%, submarine—99.0%. Its overall experimental accuracy is satisfactory, with high detection accuracy for various ship categories, except for a relatively lower accuracy in detecting destroyers. The limitations in the data scale and the impact of slicing size during sample creation may result in suboptimal classification performance for large vessels such as destroyers.Due to constraints in the slice intervals and sizes, some slices of large vessels in the training set may not be sufficiently complete, affecting model training and leading to a suboptimal classification performance.

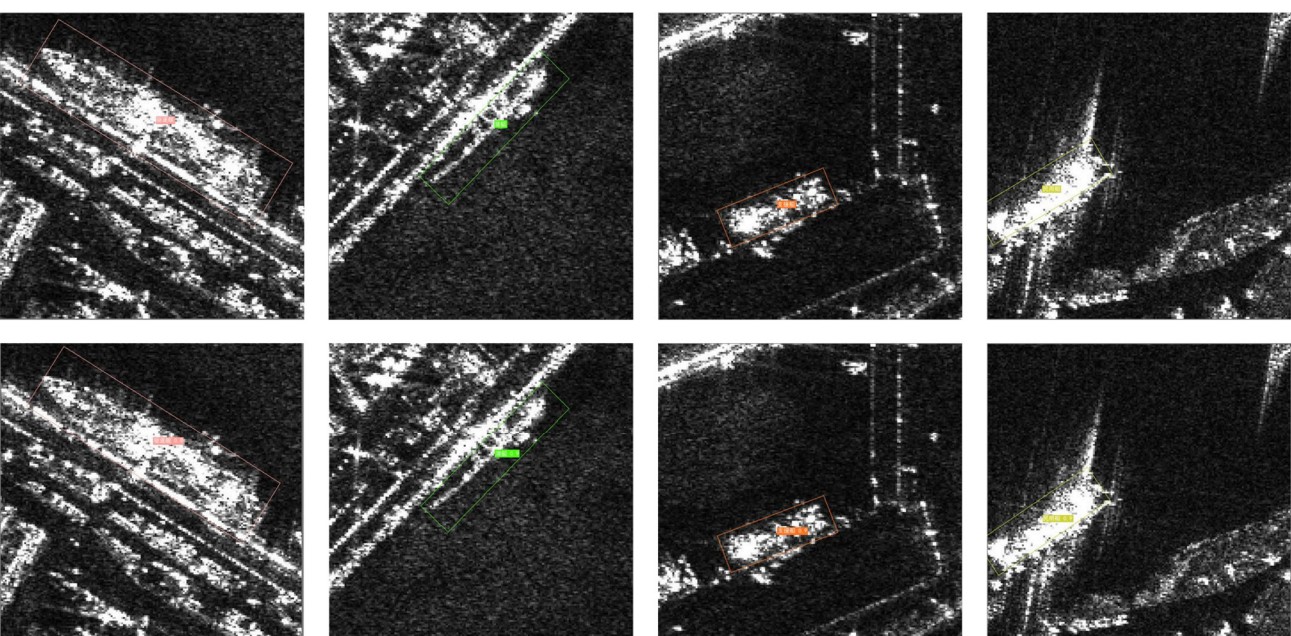

**Figure 10.** Verification experiment. The first row represents the ground truth, and the second row represents the detection results. From left to right, the categories of ships correspond to destroyer, submarine, support ship, and general ships (civilian, research, etc.).

**Table 4.** Object detection $AP_{50}$ Scores by category.

| Module/Metric | $AP_{50}$ |
|---|---|
| All | 0.942 |
| Destroyer | 0.737 |
| Support ship | 0.944 |
| Ship (civilian, research, etc.) | 1.000 |
| Submarine | 0.990 |

## 5. Discussion

In this study, we proposed a lightweight ship detection model for SAR images based on YOLOv7, with rotated bounding boxes. We enhanced the YOLOv7 model and conducted relevant experiments. The results in Sections 4.2 and 4.3 demonstrate the excellent performance of the proposed model on the validation dataset. Furthermore, the model outperforms existing rotated bounding box detection models in terms of evaluation metrics and visual aspects. Additionally, we discussed the experimental results in conjunction with a theoretical analysis.

The Kullback–Leibler Divergence (KLD) Loss Function: In order to enhance localization accuracy and detection effectiveness, and to address the issues in previous rotated bounding box detection methods such as abrupt changes in boundary angles, square-like detection problems, and inconsistencies in loss and evaluation, we introduced the KLD loss function. As is evident from Table 2 and Figure 9, the utilization of the KLD loss function during training eliminates the occurrence of square-like bounding boxes observed in the Gliding Vertex model, resulting in more accurate localization and improved precision.

The Impact of the BBCS Module: The BBCS module employs dynamic sparse queries on shallow feature maps, filtering out irrelevant key–value pairs at a coarse-grained level, retaining a small set of key routing regions and enhancing small target detection by the fine-grained querying of key routing regions. As shown in Table 1, this approach effectively improves the model's performance metrics. The visual results in Figures 8 and 9 demonstrate that this mechanism significantly aids in small target detection.

The Impact of the P-ELAN Module: The PConv ELAN (P-ELAN) module reduces the number of channels involved in computations during the convolution process, resulting in a reduction in model parameters and computational load. As indicated in Table 1, replacing this module allows for a significant reduction in model parameters and computational load without the need for complex sparsity and distillation training, while maintaining a certain level of accuracy. An examination of Table 1 and the visual results in Figure 8 show a notable decrease in model parameters and computational load.

Comparison with Other Models: Overall, our model demonstrates an outstanding detection performance, as is evident from both Table 2 and Figure 9. No instances of missed detections, as observed in models like ROI-Transformer, Rotated FCOS, and Rotated Faster R-CNN, are present. Square-like bounding box occurrences, as observed in the Gliding Vertex model, are also absent. Although Redet and Rotated RetinaNet show satisfactory detection results, our model surpasses them in terms of accuracy, model size, computational load, and overall detection effectiveness.

Limitations of this method: Firstly, our model still struggles to accurately detect ships in overly complex scenarios, necessitating further optimization of the model and its training data. Secondly, the process of creating slices for the test dataset in this study needs improvement, as the small size of the slices results in incomplete representation of large ships in the samples, ultimately making it challenging for the model to accurately detect large ships.

## 6. Conclusions

This paper introduces a rotation box object detection model based on the YOLOv7 framework (YOLOv7oSAR). The model has improved localization accuracy and detection precision through its incorporation of rotation box detection and the Kullback–Leibler Divergence (KLD) loss function. Additionally, the model includes the BRA attention mechanism to enhance its small object detection capabilities. By adjusting the model's width and depth and introducing the lightweight P-ELAN structure, the model reduces its parameters and computational resources, making it suitable for deployment. The model achieves state-of-the-art detection results on the RSDD and demonstrates excellent detection performance on a custom classification dataset. Since the image resolutions of the RSDD and the custom dataset range from 3 to 25 m, with the custom dataset having a resolution of 1 m, this highlights the model's outstanding performance across various SAR image

datasets with resolutions ranging from 1 m to 25 m, showcasing its robustness. However, despite the model's outstanding performance in detection, there is room for improvement in its handling of complex scenarios near the shore. In terms of detection validation, the model exhibits accurate results, but due to limitations in the scale of the custom dataset, larger datasets are needed to refine the dataset creation process for comprehensive testing and subsequent algorithm improvements. Additionally, there is a scarcity of datasets specifically designed for classifying rotated bounding boxes in ship detection based on SAR images. This study's future goal is to establish a larger-scale dataset, test the improved model developed in this study, and further enhance the model's ship detection capabilities in complex scenarios near the shore.

**Author Contributions:** Validation, E.S.; formal analysis, Y.M.; Investigation, W.Y.; resources, S.Z.; writing—original draft, Y.L.; project administration, F.C.; funding acquisition, J.Y. All authors have read and agreed to the published version of the manuscript.

**Funding:** This work was funded by the National Natural Science Foundation of China (Grant No. 42201063), the Innovation Driven Development Spe-cial Project of Guangxi (GuikeAA20302022), the High-Resolution Earth Obser-vation System (Grant No. "30-Y60B01-9003-22/23"), the Key Research and Development Program of Hainan Province (ZDYF2021SHFZ260) and the Hainan Provincial Natural Science Foundation of China (322QN345).

**Data Availability Statement:** The data of the experimental images used to support the findings of this research are available from the corresponding author upon reasonable request. The data are not publicly available due to privacy.

**Acknowledgments:** We thank for Wang [16], Zhu [37], and Chen [38] sharing their coding. We are also thankful for the data from the remote sensing satellite ground station of High-Resolution Satellite 3. Thank you for providing satellite imagery data, Planet Labs. We express our sincere gratitude for the valuable information and support that these data have provided to our research.

**Conflicts of Interest:** The authors declare no conflicts of interest.

## Abbreviations

| | |
|---|---|
| KLD | Kullback–Leibler Divergence |
| SAR | Synthetic aperture radar |
| CBS | Convolutional-Batch Normalization-Silu |
| BRA | Bi-Level Routing Attention |
| BCBS | Bi-Level Routing Attention Convolutional-Batch Normalization-Silu |
| ELAN | Efficient Layer Aggregation Networks |
| P-ELAN | PConv ELAN |
| FLOPs | Floating-point operations |
| CFAR | Constant False Alarm Rate |
| RCNN | Region-based Convolutional Neural Network |
| SSD | Single-Shot MultiBox Detector |
| YOLO | You Only Look Once |
| RSDD | Rotated Ship Detection Dataset in SAR Images |
| VI-CFAR | Variable Interval Constant False Alarm Rate |
| VTM-CFAR | Variable Threshold Method Constant False Alarm Rate. |
| OS-CFAR | Ordered Statistic Constant False Alarm Rate |
| CMLD-CFAR | Constant False Alarm Rate Cell Averaging Mean Level Detector |
| ARPN | Attention Receptive Pyramid Network |
| RFB | Receptive field modules |
| CBAM | Convolutional block attention modules |
| IOU | Intersection over Union |
| Conv | Convolution |
| PConv | Partial convolution |
| DWConv | Depthwise Separable Convolution |
| Gconv | Group Convolution |

|       |                                                 |
|-------|-------------------------------------------------|
| RoI   | Region of Interest                              |
| ReDet | Rotation-equivariant detector                   |
| FCOS  | Fully Convolutional One-Stage Object Detection  |
| MP    | Max-Pooling                                      |
| REP   | Reparameterization                              |
| BN    | Batch normalization                             |
| Cat   | Concatenate                                      |
| SL    | Sliding Beamforming                             |
| SPPCSPC | Spatial Pyramid Pooling Cross Stage Partial Concatenate |

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
