# Peer review of "YOLOv7oSAR: A Lightweight High-Precision Ship Detection Model for SAR Images Based on the YOLOv7 Algorithm"

_remotesensing, doi:10.3390/rs16050913_

Round 1
Reviewer 1 Report
Comments and Suggestions for Authors
This paper proposed a rotation box detection mechanism YOLOv7oSAR from SAR image based on YOLOv7 framework to address the challenges of low accuracy, imprecise localization of small targets, and deployment complexity posed by traditional rotation box algorithms. Experiments based on RSDD and verification dataset shows high overall detection accuracy.
My suggestions are listed as follows.
(1) P1, L34, SAR satellites around the world should be introduced, instead of only mentioning China's small satellites.
(2) P4, in the first paragraph of section 2, the introduction about the proposed method is too coarse and more details are suggested.
(3) In the manuscript some figures such as Figure 2, 3, etc. are not clear, and formulas are not well laid out. Literature citations should be standardized.
(4) P12, the title of Section 3 is inappropriate.
(5) P13, L515,what is the meaning of multi-view processing? Maybe it should be multi-look processing. What is the meaning of channel expansion?
(6) P17, the title of table 2 is not proper.
(7) P17,more details about the verification dataset should be provided.
(8) Can authors make some discussions about what resolution SAR images the proposed YOLOv70SAR is suitable for?
Reviewer 2 Report
Comments and Suggestions for Authors
Authors introduce a novel rotation box object detection model called YOLOv7oSAR, aiming to address the challenges of low accuracy, imprecise localization of small targets, and the hindrance to deployment posed by the complexity in current rotation box algorithms. The results of the article are useful for science and practice. However, the following points should be explained or modified:
1.Author mentioned YOLOv7 is the latest version (line 198). However, the release time of YOLOv8 is later than YOLOv7, so this description is inappropriate.
2. Since the latest version is the reason for choosing YOLOv7, why not choose YOLOv8? In addition, since YOLOv7 is no longer the latest version, why did the author choose YOLOv7 instead of YOLOv5 or YOLOv8? What are the advantages of YOLOv7? Please explain.
3. The variables in section 2.2, such as Cx, Cy, h, etc., should be italicized. variables in other places should also be modified.
4. I'm afraid I don't know which part of the text describes Figures 2,3,5,6 and 10, as they are not clearly labeled in the article.
5. Table 2 needs a title.
6.Author has compared their method with several advanced methods, and their method has demonstrated excellent performance. Additionally, I am still quite interested in comparing the results obtained by applying this algorithm to other YOLO algorithms, like YOLOv5 and YOLOv8, with those of YOLOv7. I am not sure if author has conducted similar experiments.
Reviewer 3 Report
Comments and Suggestions for Authors
Dear Authors,
Thank you for your work.
The state-of-the-art and the objective are defined, but their explanation is approximative (following comments). The method presents the theoretical aspects in its main parts, and the experiments you gave seem to confirm the efficiency of the proposed method.
You should perform the following actions.
1. Complete the list of the abbreviations (e.g., "Conv" does not appear in the list)
2. provide a DOI for each cited paper (at least, a URI to get it)
3. Clarify where the dataset is downloadable
4. provide a source of your code and a complete set of instructions to replicate the experiment
5. What evidence would you like to put in figure 10?
6. Reference all the figures in the text and explain them
7. Please provide a complete description of the computational complexity for the entire algorithm (extend the observation at line 338 for the rest of the method)
Best Regards
Round 2
Reviewer 2 Report
Comments and Suggestions for Authors
Author has made revisions to the article based on the comments, so I agree to accept this paper in present form.
Reviewer 3 Report
Comments and Suggestions for Authors
Dear Author,
The paper is accepted.
Best Regards